# Incidence of the Risk of Malnutrition and Excess Fat Mass, and Gait Speed as Independent Associated Factors in Community-Dwelling Older Adults

**DOI:** 10.3390/nu15204419

**Published:** 2023-10-18

**Authors:** Miriam T. López-Teros, Helen J. Vidaña-Espinoza, Julián Esparza-Romero, Oscar Rosas-Carrasco, Armando Luna-López, Heliodoro Alemán-Mateo

**Affiliations:** 1Centro de Evaluación del Adulto Mayor, Departamento de Salud, Universidad Iberoamericana Ciudad de México, Prolongación Paseo de Reforma 880, Lomas de Santa Fe, Ciudad de México 01219, Mexico; miriam.lopez@ibero.mx (M.T.L.-T.);; 2Coordinación de Nutrición, Centro de Investigación en Alimentación y Desarrollo (CIAD), A.C., Carretera Gustavo Enrique Astiazarán Rosas, No. 46, Col. La Victoria, Hermosillo 83304, Sonora, Mexico; 3Dirección de Investigación, Instituto Nacional de Geriatría, Ciudad de México 10200, Mexico; aluna@inger.gob.mx

**Keywords:** nutrition disorders, risk factors, body composition, physical functional performance, independent living, and aged

## Abstract

Background and aims: Only one cohort study exists on the incidence of the risk of malnutrition (RM) in older adults, though numerous cross-sectional reports, identified several risk factors associated with the prevalence and incidence of this condition. However, alterations in body composition and impaired physical performance as exposition variables of RM have not been explored. This study assessed the incidence of RM and determined its association with excess fat mass, low total lean tissue, gait speed, and handgrip strength as exposition variables for RM in community-dwelling older adults. Methods: This is a secondary analysis of older adults (≥60 years) derived from the study “Frailty, dynapenia, and sarcopenia in Mexican adults (FraDySMex)”, a prospective cohort project conducted from 2014 to 2019 in Mexico City. At baseline, volunteers underwent body composition analysis and physical performance tests. Several covariates were identified through comprehensive geriatric assessment. At baseline and follow-up, RM was assessed using the long form of the mini nutritional assessment (MNA-LF) scale. Associations between the exposition variables and RM were assessed by multiple logistic regression. Results: The cohort included 241 subjects. The average age was 75.6 ± 7.8 years, and 83.4% were women. The mean follow-up period was 4.1 years, during which 28.6% of subjects developed RM. This condition was less likely to occur in those with an excess fat mass, even after adjusting for several covariates. Regarding total lean tissue, the unadjusted model showed that RM was more likely to occur in men and women with a low TLT by the TLTI classification, compared to the normal group. However, after adjusting for several covariates (models 1 and 2), the association lost significance. Results on the association between gait speed and RM showed that this condition was also more likely to occur in subjects with low gait speed, according to both the unadjusted and adjusted models. Similar results were found for RM in relation to low handgrip strength; however, after adjusting for the associated covariates, models 1 and 2 no longer reached the level of significance. Conclusions: RM diagnosed by MNA-LF was significantly less likely to occur among subjects with excess fat mass, and a significant association emerged between low gait speed and RM after 4.1 years of follow-up in these community-dwelling older adults. These results confirm the association between some alterations of body composition and impaired physical performance with the risk of malnutrition and highlight that excess fat mass and low gait speed precede the risk of malnutrition, not vice versa.

## 1. Introduction

Older adult populations are more susceptible to developing risk of malnutrition (RM). Current pooled analytical data on RM in older subjects highlights this condition as a serious public health problem in aged populations [1]. That analysis reported a prevalence of 46.2% in the geriatric population in different settings on five continents (mostly European countries) and 31.9% among community-dwelling older adults. The latter figure is close to the extreme range of 30.5% reported for these populations [2]. High prevalence has also been reported for other non-Caucasian, community-dwelling, older adults [3,4,5,6]. Importantly, the studies included in the pooled data analysis and other research on the prevalence of RM used the long form of the mini nutritional assessment (MNA-LF) scale to assess RM. This scale, generated and validated in older adults [7], has shown sensitivity and specificity of 96% and 98%, respectively, for diagnosing RM, relative to conventional nutritional assessments and clinical diagnoses [8].

In Mexico, prevalences of RM in community-dwelling older adults ranging from 30.3 to 50%, assessed by the MNA-LF, have been reported in cross-sectional studies based on either probabilistic [9] or non-probabilistic sampling [10,11,12,13,14]. A recent study of a national-level sample of older adults reported a prevalence of 40.4% using the modified short version of the MNA [15]. All these studies show high prevalences of RM in community-dwelling older Mexican adults. Regarding incidence, a cohort study of older Swedish adults found RM rates of 8.2, 16.1, 10.8, and 8.2% at 1, 2, 3, and 4-year follow-ups, respectively [16]. Overall, RM now appears to constitute a public health challenge due to its current high prevalence and the likelihood that it will increase in the coming years. RM is not listed in the International Disease Classification but has been independently related to numerous adverse clinical outcomes, such as loss of functionality, impaired physical performance [10,17,18], sarcopenia [19,20], and mortality [18,21]. Some related studies recognize that nutritional interventions in older adults at risk for malnutrition in distinct healthcare settings can have a positive impact on energy intake and body weight [22].

Several variables have been associated with RM diagnosed by the short form of the mini nutritional assessment, or the MNA-LF, in cross-sectional studies of community-dwelling older adults [3,4,9,10,11,12,15,23,24], one cohort study [16], and a systematic review and meta-analysis of observational studies [25]. However, the potential association between alterations of body composition, particularly excess fat mass, and low total lean tissue, on the one hand, and impaired physical performance measured by low gait speed and low handgrip strength as exposition variables, on the other, and RM has not been explored thoroughly. It is widely believed that these factors can contribute to the early development of RM through diverse mechanisms. In this context, the present study was designed to assess the incidence of RM and its association with excess fat mass and low total lean tissue, low gait speed, and low handgrip strength as exposition variables in community-dwelling older adults after 4.1 years of follow-up.

## 2. Materials and Methods

### 2.1. Data Sources and Study Design

This is a secondary analysis based on the primary data generated in the project “Frailty, dynapenia, and sarcopenia in Mexican adults” (FraDySMex), a prospective cohort study of a non-representative sample of community-dwelling men and women subjects aged ≥50, conducted from 2014 to 2019. The aim of the original project was to assess body composition by dual-energy X-ray absorptiometry (DXA) and physical performance using the short physical performance battery and measures of muscle strength by dynamometry, to diagnose osteoporosis, dynapenia, frailty, and sarcopenia. In addition to the body composition and physical performance measures, volunteers underwent a health and nutritional status evaluation and other functional measurements. Social and demographic characteristics at baseline were recorded as part of a comprehensive geriatric assessment (CGA). The protocol is in full accordance with the ethical standards established in the 1964 Helsinki Declaration with amendments. The study protocol was reviewed and approved by the Ethics Committee of the Angeles Mocel General Hospital and registered at Mexico’s National Institute of Geriatrics (DI-PI-002/2014). All subjects signed an informed consent form.

The methodology for selecting the study population was published previously [13] [26], so here we provide only a brief description. Baseline data were collected from October 2014 to December 2015, while follow-up data were gathered from October to December 2019. At baseline, 540 potential men and women subjects were included. All procedures were performed in the Functional Evaluation Research Laboratory at Mexico’s National Institute of Geriatrics and the Older Adult Evaluation Center of the Universidad Iberoamericana (Mexico City), where subjects underwent the aforementioned series of objective evaluations, performed by a medical team made up of a physician, a nutritionist, a psychologist, physical therapists, physical rehabilitators, social workers, and geriatricians, all previously and fully informed of the study’s aims.

### 2.2. Study Population

All participants were community-dwelling adults and older adults from two municipalities (of a total of 16) in southeast Mexico City (Magdalena Contreras, and Álvaro Obregón), characterized by high poverty levels and older adult populations. Subjects were invited regardless of their health, mental, nutritional, and functional status via home visits by a psychologist or social worker, or through flyers left in churches, seniors’ community centers, social security centers, and health clinics in the two municipalities.

No institutionalized individuals or potential subjects with an acute or chronic disease that could affect the measurements required by the study protocol were invited. Potential volunteers had to reach the installations or laboratories on their own. Upon arrival, they underwent a cognitive assessment, and the mini-mental state examination (MMSE) was applied. Subjects had to be able to answer the study questionnaires by themselves. If they had MMSE scores of 10 points or less [27], a caregiver assisted them. All subjects included at baseline with complete data were contacted by phone and home visits at follow-up and invited to participate in the second assessment.

The inclusion criteria for the present analysis were men and women subjects ≥60 years old. At baseline and follow-up, they had to have the complete corpus of measurements required to test the hypothesis variables, as well as RM, assessed by the MNA-LF, and all data collected in the CGA at baseline, including the cognitive function assessments, notes on depressive symptoms, comorbidity, smoking and alcohol consumption, and the number of medications taken. Functional status was assessed by the basic activities of daily living (BADL) and instrumental activities of daily living (IADL) scales. Evaluations of physical performance were based on gait speed and handgrip strength. Social and demographic characteristics were assessed, and body composition by DXA was included. The exclusion criteria before follow-up were potential subjects with incomplete files, individuals <60 years old, and diagnoses of RM at baseline by the MNA-LF.

### 2.3. Risk of Malnutrition as the Main Response Variable

At both baseline and follow-up, RM was assessed by the MNA-LF scale [7], applied by trained personnel following the instructions in the MNA^®^ User Guide (www.mna-elderly.com, accessed on 1 October 2014). To complete the MNA, a face-to-face interview was conducted, where volunteers were asked about their lifestyle, medications, mobility, signs of depression or dementia, the number of meals taken per day, food and fluid intake, autonomy of feeding, and self-perceptions of their health and nutrition. They also underwent an anthropometric assessment since the MNA-LF requires data on body weight, height, and arm and calf circumference measurements. Based on the questions answered on the scale, scores were obtained, and nutritional status was classified according to their MNA-LF scores: <17, malnutrition or undernutrition; 17–23.5, risk of malnutrition; and 24–30, normal nutritional status [7]. RM, as the main response variable or outcome, was coded as the categorical variable. For this analysis, the MNA-LF was dichotomized into well-nourished (24–30 points) and at risk for malnutrition (≤23.5 points). In the latter category, nine individuals had mal- or undernutrition (<17 points) and, hence, were included in the analysis.

### 2.4. Anthropometry

The anthropometric variables of body weight and height were measured to the nearest 0.1 kg and 0.1 cm, respectively, using a SECA mBCA 514 scale (MFBIA; SECA^®^, Hamburg, Germany) and a SECA 264 Free-Standing Wireless 360 Stadiometer (SECA^®^, Hamburg, Germany). BMI was calculated with these measurements. Arm and calf circumferences were measured by the ISAK technique. The BMI and arm and calf circumferences were incorporated into the MNA-LF. The BMI was also used to diagnose overweight and obesity [28]. All variables were measured by trained personnel as part of the anthropometry protocol.

### 2.5. Body Composition Measured by DXA

At baseline, whole and regional body composition was measured in fasting conditions by DXA (Hologic Discovery-WI, Hologic Inc., Bedford, MA, USA). Volunteers wore disposable gowns. All metallic objects were removed. Measurements and calibrations of the DXA system were carried out following the manufacturer’s instructions. All the DXA scans were edited to estimate appendicular lean tissue following published recommendations [29]. For the present analysis, only total lean tissue (TLT), bone mineral content (BMC), and fat mass (FM) were considered. The TLT and FM results were divided by height in meters^2^ to obtain the TLT and FM indices (TLTI, kg/m^2^ and FMI, kg/m^2^, respectively). These indices were then used to derive the exposition variables. In addition, published FMI cut-off points were used to diagnose excess fat mass in the whole sample [30].

### 2.6. Physical Performance Assessment

Gait speed and handgrip strength were the main components of the physical performance assessment at baseline. Results were used to derive the exposition variables. Both tests formed part of the CGA. After receiving instructions, volunteers performed the gait speed test, recorded as a regular 6-m walk on the GAIT Ritel instrumented mat (platinum 20,204 × 35.5 × 0.25 inches, 100 Hz sampling rate). Results were recorded in meters/second (m/s). Isometric grip strength was measured in both hands with an adjustable handgrip strength dynamometer (JAMAR Hydraulic Hand Dynamometer, Lafayette, IN, USA), following the standardized, approved protocol [31,32]. Results were recorded in kilograms (kg). The mean of three trials for each hand was recorded, but only the highest value was used for the ensuing analyses.

### 2.7. Association between the Exposition Variables and RM

The total lean tissue index and fat mass are the main body composition compartments associated with RM. With certain other factors, these two factors were recently related to RM in older people. The results show that RM is less likely to occur in subjects <85 years old, those with normal swallowing and gait speed, and higher TLTI, fat mass, and BMI values, while those ˃85 had a greater likelihood of suffering RM, according to adjusted and unadjusted models [24]. One cohort study showed that RM was less likely to occur in people with higher triceps skinfold thickness, greater handgrip strength, and better overall physical health, but that risk increased with greater age and low serum albumin at baseline [16]. For our analysis, alterations of the body composition compartments, particularly low TLT and excess fat mass, were considered separately as exposition variables. The former was categorized as normal or low, based on sex-specific cut-off points. The dividing point was a value <20th percentile of the data distribution of the TLTI of the study population as recommended by EWGSOP2 [33]. Excess fat mass was determined by the sex-specific FMI (kg/m^2^) cut-off points reported by Kelly et al. (2009) [30]. Regarding the components of impaired physical performance as exposition variables of RM, we considered gait speed (m/s) and handgrip strength (kg), two aspects recently associated with RM, among other variables. RM was less likely to occur in older adults with normal gait speed (≥8 m/s) [24] and in those with greater handgrip strength at baseline [16]. For our study, normal gait speed for older men and women was considered ≥0.8 and ≥0.7 m/s, respectively, so low gait speed for men was ≤0.8 m/s, while for women it was ≤0.7 m/s. Handgrip strength was categorized as normal at scores of >18 kg and >10 kg for men and women, respectively, so low handgrip strength was ≤18 kg in men and ≤10 kg in women. These cut-off points were derived from the distribution of the variables in the same study population and correspond to values <20th percentile as recommended by EWGSOP2 [33]. These exposition variables were coded as categorical (Table 1).

### 2.8. Assessment of Covariates

The variables used as covariates were evaluated as part of the CGA. A questionnaire was applied to obtain sociodemographic information on subjects’ age, occupation, marital status, years of schooling, and medical services. Based on the occupational data, the main source of income was classified as “pension”. Information on living arrangements was coded as living alone or not. Age, gender, educational level, living alone, being single, widowed, or divorced, not having a pension, and health insurance status are among the factors that have been associated with RM in older people in earlier studies [4,9,15,25].

Cognitive status was assessed and classified using the MMSE validated for Spanish-speaking populations [27]. Cognitive decline or impairment was considered at MMSE scores ≤23 points for subjects with 5 years of schooling, ≤19 for those with 1–4 years, and ≤16 for those with no schooling or <1 year. Depressive symptoms were assessed by the Center for Epidemiologic Studies Depression Scale, Brief Version (CESD-7), validated for older Mexican adults [34]. The presence of these symptoms was considered when subjects scored 5 or more points. Cognitive status [23,35], depression, and depressive symptoms have all been associated with RM assessed by the MNA in older people [3,9,12,16].

Comorbidity was assessed by Charlson’s comorbidity scale [36], which was applied by trained personnel following published instructions and recommendations [37]. Comorbidity was coded as no (≤2 diseases) or yes (≥3 diseases) [13]. Polypharmacy was assessed by asking subjects for the names of their medications, their pharmaceutical form, dosage frequency, duration of use, and if they were prescribed medically. Polypharmacy was defined as yes if 5 or more medications were being taken and no if the number was ≤4 per day [38].

Data on current or previous smoking, frequency of alcohol consumption, and use of drugs were also assessed and coded as follows: Alcohol consumption if subjects drank ≥2 glasses/day [39] and current smoking as yes or no [40]. Comorbidities and smoking have been associated with RM in community-dwelling older adults [4], while polypharmacy has been signaled in cases of individuals living under conditions of home care [35]. In addition, oral health (number of dental pieces, dry mouth, and xerostomia) has been related to RM [23,41], as have poor self-perceptions of health [15,16].

Functional status was assessed by Barthel’s scale for BADLs [42] and Lawton and Brody’s scale for IADLs [43]. Functional dependence was present if volunteers had scores of ≤90 or ≥1 on these two instruments, respectively. Loss of functionality has been associated with RM in older people [9,10,23], whereas both handgrip strength and healthy gait speed as continuous variables are deemed to be protector factors from RM in terms of the physical performance components of community-dwelling older adults [16,24].

### 2.9. Statistical Analysis

The student’s *t*-test for continuous variables and chi-squared test for categorical variables were used to detect significant between-group differences with and without RM (*p*-value ≤ 0.05). Results are reported as mean values with standard deviation (SD) or percentages (%), respectively. A simple logistic regression analysis was used to test for potential associations between several of the independent variables and RM as the response variable (Table 2). Two criteria were established to determine a potential association: A *p*-value ≤ 0.2 and a biological plausibility of the odds ratio (OR). Multiple logistic regression using automated stepwise analysis (beginning with forward using a *p*-value ≤ 0.05) was applied to test the association between each exposition variable and the occurrence or development of RM. It is important to note that the regression models were generated separately due to the collinearity between the exposition variables and several covariates. Once obtained, preliminary models were assessed for an interaction at a *p*-value of ≤0.1. Collinearity was assessed using a correlation matrix (r ≤ 0.85). The linearity assumption of the model was not assessed because the only categorical variables were included. The OR and 95% confidence interval (CI) for the occurrence of RM were calculated using the variables of body composition alterations and impaired physical performance, after adjusting for several covariates in the final models. All analyses were performed with the statistical package STATA 16.0 for Windows (Stata Corp, College Station, TX, USA).

## 3. Results

### 3.1. Baseline Characteristics

At baseline, 540 men and women completed all the required data, but 40 subjects <60 years of age were excluded. According to their MNA-LF scores, 22.9% had RM, with women having a higher proportion than men (84 vs. 16%). None of the subjects diagnosed with RM by the MNA-LF at baseline were included in the follow-up study (*n* = 100), so the potential sample consisted of 400 men and women. Some volunteers, however, had incomplete MNA-LF data (*n* = 55). Other losses occurred during follow-up due to four main causes: Excluded candidates (*n* = 39), unable to contact due to change of address (*n* = 16), failure to attend the appointment (*n* = 6), and death (*n* = 43). This left a total sample for analysis at baseline and follow-up of 241 men and women. Figure 1 shows the flow chart of volunteers throughout the cohort study.

At baseline, the sample of 241 men and women subjects (83.4% women) had an average of 75.6 years and well-nourished status according to the MNA-LF. In addition, 85.4% of the sample had good health by self-perception, with only 14.5% reporting poor health. Regarding fat mass measured by DXA, 85.4% had excess fat mass according to the FMI classification, while 21% had low TLTI, 36.2% had comorbidities, and 85.4% had overweight and obesity, estimated by BMI ≥ 30 kg/m^2^. Regarding physical performance and functional status, 33.2% had low gait speed, 37.7% low handgrip strength, and 14.1% and 32.3%, respectively, with dependency according to the BADL and IADL scales.

Table 1 shows the behavior of several variables according to the four exposition variables measured at baseline. Regarding the behavior of these variables according to the fat mass index classification taken as the dichotomic variable, subjects with excess fat mass were younger than the normal group. Mean values for body weight, BMI, fat mass, TLT, and BMC were all higher than those for the normal fat mass group (*p* ≤ 0.05). In line with these findings, the MNA score was significantly higher than in the normal fat mass group. The low TLT group according to the TLTI classification was also younger; nevertheless, they had significantly lower mean values of body weight, BMI, fat mass, TLT, and BMC. In addition, the mean value of the MNA score was significantly lower than in the normal TLT group. In contrast, the low gait speed group was older, and the proportion of subjects with depression symptoms and dependency, measured by IADL, was higher (*p* ≤ 0.05). Concurring with these results, they also had significantly lower mean HGS values than the normal gait speed group. The MNA score was also lower than in the normal gait speed group. Finally, the low HGS group was also older than the normal group and had a higher proportion of subjects with dependency by IADL, accompanied by a lower mean MNA score than the normal HGS group.

The mean follow-up period was 4.1 years. In that interval, 28.6% of the sample developed RM. The cumulative incidence was more pronounced in women than men (84.1 vs. 15.9%). Regarding the factor of incidence according to the exposition variables, Table 2 shows that a lower proportion of older adults (25.2%) with excess fat mass by FMI classification developed RM compared to the normal group (48.5%) at 4.1 years of follow-up. In another result, 41.1% of the subjects with low TLT by TLTI classification had developed RM at follow-up, compared to just 24.7% of the normal TLT group. Similar findings emerged when the incidence of RM was compared between the low and normal gait speed and handgrip strength groups. Overall, these results suggest that older subjects with excess fat mass presented lower incidences of RM, while those with low TLT, low gait speed, and low HGS had higher incidences compared to the subjects without these conditions at 4.1 years of follow-up. Table 3 shows the relative change at 4.1 years of follow-up in several variables, such as age in years, anthropometry, body composition, nutritional status, and physical performance. Significant relative changes were found for most of the variables, except total lean tissue, TLTI, BMC, and HGS.

Table 4 shows the potential associations between our hypotheses and several independent variables and RM as the response variable. Results of the simple logistic regression show that excess fat mass, low TLT, low gait speed, low HGS, marital status, education, good self-perceptions of health, depression symptoms, cognitive impairment, comorbidity, polypharmacy, dry mouth, and dependency by the BADL and IADL were all associated with RM estimated by the MNA-LF.

### 3.2. Final Models of the Association between the Exposition Variables and RM at 4.1 Years of Follow-Up

Regarding the association between alterations of body composition and the response variables, results show that the RM was less likely to occur in subjects with excess fat mass by the FMI classification. The unadjusted model had an OR of 0.35 [95% CI: 0.17–0.74]. This result remained significant after adjusting for BADL dependency, low gait speed, and good self-perceptions of health (model 1; OR: 0.24 [95% CI: 0.09–0.64]), and even for IADL dependency, low gait speed, and good self-perceptions of health (model 2; OR: 0.35 [95% CI: 0.16–0.82]) compared to the group without excess fat mass or normal fat mass group (Table 5). In contrast to its relation to excess fat mass, RM was more likely to occur in men and women with low TLT by the TLTI classification, compared to the normal TLT group. The unadjusted model had an OR of 2.30 [95% CI: 1.21–4.39]). This finding remained at risk but lost significance after adjusting for BADL dependency, dry mouth, and self-perceptions of health (model 1; OR: 2.06 [95% CI: 0.98–4.32]). The same results were obtained after adjusting for IADL dependency, dry mouth, gait speed, and self-perceptions of health (model 2; OR: 2.04 [95% CI: 0.97–4.26]).

For the association between aspects of impaired physical performance, such as low gait speed and the response variable, Table 5 shows that RM was more likely to occur in men and women with low gait speed, according to the unadjusted model (OR: 2.43 [95% CI: 1.36–4.32]). This result remained significant after adjusting for dry mouth, excess fat mass, and self-perceptions of health (model 1; OR: 2.01 [95% CI: 1.06–3.83], and even for dry mouth, the TLT, and self-perceptions of health (model 2; OR: 1.96 [95% CI: 1.03–3.72]. Similar results were found for low handgrip strength, though in this case, both models 1 and 2 lost significance after adjusting for the associated covariates.

## 4. Discussion

All subjects ≥60 who were diagnosed with RM by the MNA-LF scores at baseline were excluded from the analysis. A significant proportion of the sample had excess fat mass according to the FMI classification ranges [30], and overweight and obesity by the BMI categories [28]. Around 29% (mostly women) developed RM at follow-up. These data on the incidence of RM enhance our understanding of the etiology of this condition in community-dwelling older adults by providing a basis for estimating the risk of developing malnutrition, as well as some other diseases. Moreover, the unadjusted and adjusted models that incorporated several covariates showed that RM was less likely to occur in subjects with excess fat mass diagnosed by the FMI classification ranges, and more likely to occur in subjects with low TLT diagnosed by the TLTI classification and low gait speed at follow-up. None of these results have been reported previously for groups of community-dwelling older people worldwide.

Based on the clinical consequences reported in the scientific literature, the volunteers diagnosed with RM are likely at a greater risk of functional dependency [10,17], sarcopenia [18,19], and mortality [20,21] than well-nourished older subjects. From a public health perspective, the diagnosis and early confirmation of low gait speed should be warning signals of the need to take action to treat the people so affected and prevent further progress or evolution of RM and its numerous adverse clinical consequences in community-dwelling older adults, especially those in low- and middle-income countries. It seems mandatory to assess body composition and physical performance, among other factors, to detect RM early in older adult populations (Table 1 and Table 5).

### 4.1. Previous Studies Related to the Causality of Risk of Malnutrition

Our search identified only one cohort study on the association between several risk factors and RM. That published work reported that high age, low self-perceptions of health, and depressive symptoms were factors strongly associated with RM at follow-up [16]. The results of several published cross-sectional studies [3,4,9,10,11,12,15,23,24] and a recently published systematic review and meta-analysis of cross-sectional studies [44] on the association between risk factors and RM in older populations led us to hypothesize that excess fat mass, low TLT, low gait speed, and low handgrip strength, separately, could be associated with RM at follow-up in community-dwelling older people. Results of the present cohort study also found that RM is less likely to occur in subjects with excess fat mass (measured by DXA and diagnosed by the FMI classification) assessed as the categorical variable. These results are in line with those published in a cross-sectional study on the association between fat mass by DXA and RM [24] and triceps skinfold measurements, on the one hand, and RM at baseline, on the other, in a cohort of older adults [16]. It seems that RM is also less likely to occur in subjects with excess fat mass, but that higher fat mass index and fat mass values increase the risk of impaired physical performance [45] and greater physical dependency [46], respectively, in older adults. Finally, we cannot omit the fact that high fat mass is related to several cardiovascular disease outcomes [47] and mortality [48]. At this time, more cohort studies are required to support or reject the results reported herein.

Regarding the association between low TLT and RM, our regression models do not support previously published findings on community-dwelling older New Zealanders [24], even after considering low TLT as the categorical variable, since the researchers in that study [24] reported that RM was less likely to occur in subjects with high fat-free mass index values as the continuous variable. However, upon observing the OR values, the biological plausibility, and mainly the *p* values of the models on the association between low total lean tissue and risk of malnutrition (Table 5), we suggest that increasing the sample size could reach significance. Therefore, more studies are needed to support the findings reported in that earlier work [24].

Unfortunately, the approach adopted in our work is impractical for daily clinical practice in the study region due to the limited availability of the methods and equipment required to assess body composition. However, since many published anthropometric and BIA equations are available to accurately estimate fat mass and fat-free mass in older adults, and DXA could be an effective method for assessing fat mass and total lean tissue in older populations, it may be possible to overcome this limitation. The early measurement of fat mass and fat-free mass using ethnic- and gender-specific BIA, anthropometric equations [49,50,51], or TLT by DXA will help researchers and clinicians identify people with excess fat mass or low TLT, before this is reflected in MNA-LF scores.

Gait speed as a categorical variable has also been associated with RM, as this condition was less likely to occur in community-dwelling older adult subjects with healthy values (≥0.8 m/s) [24]. Kramer et al. (2022) [44] published similar findings. Results of this meta-analysis considering only cross-sectional studies, including the study published by Chatindiara et al. (2018) [24], showed that well-nourished groups walked significantly faster than those with RM. To the best of our knowledge, the present cohort study is the first to confirm this association using low gait speed as the hypothesis variable. Results showed that RM was more likely to occur in subjects with low gait speed even after adjusting for their corresponding variables (see models 1 and 2, and Table 5). Currently, few cross-sectional studies of satisfactory quality have explored this association [44]. It is well known that the gait speed test is normally used to diagnose severe sarcopenia [33], assess physical functioning [52,53], and identify people at high risk of adverse health outcomes [54] and/or disability [55]. But it is also a simple, reproducible measure of physical performance, so future cohort studies on the associated factors and risk of malnutrition should explore this as an exposition variable to confirm or reject our findings.

Finally, only one cohort study to date has analyzed the association between several factors and RM at baseline. Those researchers found that RM was less likely to occur in subjects with high handgrip strength as a continuous variable in community-dwelling older people in Sweden. However, in the model built after follow-up, handgrip strength did not appear among the factors associated with the incidence of RM [16]. The researchers in a cross-sectional study carried out with older New Zealanders did not find any association between these two variables [24]. The meta-analysis published by Kramer et al. (2022) [44], which included the studies published by Johansson et al. (2009) [16] and Chatindiara et al. (2018) [24], showed that well-nourished groups had significantly higher HGS values than groups with RM. In the present cohort study, however, there was an association between low handgrip strength as a hypothesis variable and the incidence of RM according to the unadjusted OR, though this lost significance upon adjusting for several covariates. The results are shown in models 1 and 2 (Table 5). Upon examining the *p*-values found (models 1 and 2), it seems likely that the small sample size influenced the significance of this association. As mentioned above, 37.7% (*n* = 15) of men and women had low handgrip strength. Future cohort studies should recruit a more adequate sample size to determine if this association reaches statistical significance. Clearly, additional work is required to clarify this possible association.

### 4.2. Potential Biological Mechanisms

The association between excess fat mass and RM found in our cohort study may be supported by the following mechanisms. Triacylglycerols in adipose tissue or fat mass represent a major energy reserve in the body. Under conditions of high energy demand, such as low energy intake or caloric deficit, stored triacylglycerols are hydrolyzed by lipase and released into the bloodstream as free fatty acids to satisfy constant energy needs. In fact, there are reports that MNA scores of 17–23.5 (risk of malnutrition) detected older adult patients with poor nutritional intake and, at the same time, normal albumin levels or no loss of body weight [7]. In addition, eating only one or two meals a day [9], or consuming fewer than three snacks daily [56], increased RM compared to subjects who ate three or more meals, or more than three snacks daily. Thus, it is to be expected that RM is less likely to occur in subjects with excess fat mass since those with RM likely satisfy their energy needs, as the results of our study confirm. In contrast, total lean tissue, or fat-free mass, is associated with energy intake, an association mediated by the resting metabolic rate. Fat-free mass and the resting metabolic rate, together, explain 62% of the variance in energy intake [57]. Our results suggest that low TLT could be the main determinant of the resting metabolic rate and, hence, low energy requirements, and this condition could foster low energy intake that could contribute to the development of RM.

Finally, the relationship between nutritional status and physical performance in older adult populations is well known. Well-nourished subjects walk significantly faster than those with RM or malnourishment [44]. Impaired physical performance (e.g., low gait speed) can have an early effect on some of the activities involved in buying and preparing food, thus impacting dietary caloric intake. Gait speed correlates positively with all body mass measures, TLT, percentage of body fat, and BMI [24]. Therefore, a synergistic effect among low TLT, low fat mass, and low gait speed can be sustained. In addition, the older people in this study who developed RM at 4.1 years follow-up had less fat mass and lean tissue at baseline, compared to the men and women in the group without RM.

### 4.3. Strengths and Limitations

One of the strengths of this study is its cohort design, which allowed us to explore temporal associations and causality. Significantly, unlike the earlier cross-sectional studies and the one cohort study cited, we used excess fat mass, low TLT, low handgrip strength, and low gait speed as the exposition variables, then adjusted the models for several covariates reported in the literature but based on their tested association with RM using simple regression analysis (Table 4). In addition, the exposition variables were assessed using reference methods such as DXA for body composition and the GAIT Rite instrumented mat for gait speed. Two other strengths are that robust statistical analyses were carried out to determine associations, and the follow-up period was sufficiently long to allow us to determine associations between several exposition variables and RM.

The limitations of cohort studies in terms of losses of subjects at follow-up are well-known. However, the percentage of loss in this study was relatively low (just 12%) and was taken into account in our calculations of the sample size. We were also able to overcome this because there were no significant differences in several of the demographic, body composition, nutritional status, and clinical characteristics between the subjects who were lost and those who remained for the follow-up study (see Appendix A).

Another limitation is a possible selection bias since our sample was made up of volunteers who came to the examination centers on their own, so older adults with greater impairment of their physical or functional performance may have been excluded, possibly resulting in an underestimation of the incidence of RM and the number of people with low TLT and low gait speed and low handgrip strength. Also, our sample was made up of an overweight/obese population with a female majority and poverty. Additionally, there are many different cut-off points worldwide, and distinct approaches have been used to derive new ones. Some of the existing cut-off points are related to our hypothesis variables. In light of these published cut-off points, we recognize the importance of performing sensitivity analyses. Although this aspect was beyond the scope of the present study, we strongly recommend its implementation in future cohort studies. Finally, this study tested the association between alterations of body composition and impaired physical performance by adjusting for several covariates but did not consider biochemical and nutritional parameters like hematocrit, glucose, triglyceride, and albumin levels, or erythroid sedimentation rates, all of which have been related to RM in clinical settings.

## 5. Conclusions

The accumulated incidence of the risk of malnutrition was relatively high—around 29%—in this sample of community-dwelling older adults. RM diagnosed by MNA-LF was significantly less likely to occur among subjects with excess fat mass, and a significant association was found for low gait speed and RM at 4.1 years of follow-up in these community-dwelling older people, even after adjusting for several covariates. These results confirm the association between some alterations in body composition and impaired physical performance with risk of malnutrition. Our findings also highlight that excess fat mass and low gait speed actually precede RM, not vice versa. Future prospective cohort studies are needed to confirm or reject our hypotheses for older adult populations. Due to the high incidence of RM in community-dwelling older people, and its possible short-term clinical consequences, early assessment of body composition and physical performance to detect excess fat mass, low total lean tissue, and low gait speed, respectively, is recommended to detect the risk of malnutrition in older adults before this condition can develop.

## Figures and Tables

**Figure 1 nutrients-15-04419-f001:**
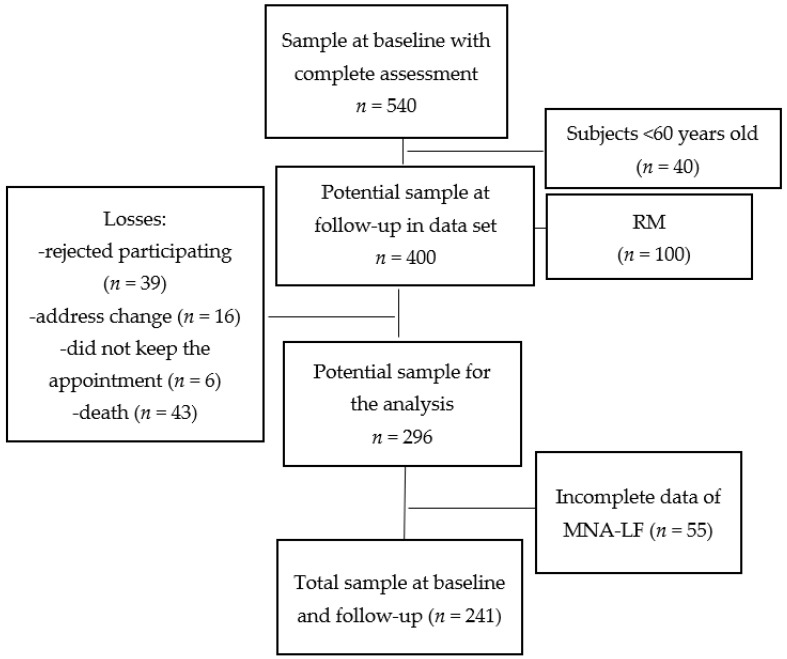
Flow chart of the volunteers during the cohort study. Abbreviations: RM = risk of malnutrition, MNA-LF = long form of the mini nutritional assessment.

**Table 1 nutrients-15-04419-t001:** Baseline anthropometric and body composition, comprehensive geriatric assessment, and sociodemographic data according to the exposition variables.

Variables	Body Composition				Physical Performance			
	Fat Mass		Total Lean Tissue		Gait Speed		Handgrip Strength	
	Excess Fat Mass ^a^ *n* = 206	Normal Fat Mass *n* = 35	Low TLT ^b^ *n* = 51	Normal TLT *n* = 190	Low GS ^c^ *n* = 80	Normal GS *n* = 161	Low HGS ^d^ *n* = 91	Normal HGS *n* = 150
Gender Women Man	171 (83.0) 35 (16.9)	30 (85.7) 5 (14.3)	39 (76.5) 12 (23.5)	162 (85.3) 28 (14.7)	64 (80.0) 16 (20.0)	137 (85.1) 24 (14.9)	75 (82.4) 16 (17.6)	126 (84.0) 24 (16.1)
Age, years	75.1 ± 7.9	79.1 ± 7.2 *	77.1 ± 8.2	79.1 ± 7.2 *	79.1 ± 7.8	73.9 ± 7.6 *	78.7 ± 7.6	73.8 ± 7.7 *
Anthropometry Height, m	1.5 ± 0.0	1.4 ± 0.0	1.5 ± 0.0	1.5 ± 0.0	1.49 ± 0.0	1.50 ± 0.0	1.49 ± 0.0	1.51 ± 0.0
Body weight, kg	60.6 ± 12.1	51.4 ± 7.6 *	57.7 ± 10.4	67.3 ± 12.8 *	65.4 ± 14.2	65.2 ± 12.3	64.1 ± 14.9	65.9 ± 11.5
BMI, kg/m^2^	29.7 ± 4.7	23.1 ± 2.8 *	24.8 ± 3.8	29.8 ± 4.8 *	29.1 ± 5.6	28.6 ± 4.8	28.5 ± 5.7	28.9 ± 4.6
Body composition Total mass by DXA, kg	65.6 ± 12.4	45.7 ± 13.2	51.1 ± 14.4	65.9 ± 12.6 *	62.7 ± 15.7	62.8 ± 13.6	61.7 ± 16.5	63.4 ± 12.8
Fat mass, kg	28.1 ± 7.2	16.2 ± 3.2 *	22.2 ± 6.2	27.6 ± 7.9 *	26.8 ± 8.8	26.4 ± 7.5	26.4 ± 9.6	26.6 ± 6.7
Fat mass index, kg/m^2^ Total lean tissue, kg	12.4 ± 3.2 35.7 ± 7.6	7.2 ± 1.3 * 31.3 ± 4.7 *	9.5 ± 2.6 29.8 ± 6.9	12.3 ± 3.5 * 36.5 ± 7.0 *	11.9 ± 3.9 35.1 ± 7.6	11.6 ± 3.3 35.1 ± 7.4	11.8 ± 4.1 34.5 ± 7.5	11.7 ± 3.1 35.5 ± 7.5
BMC, kg	1.7 ± 0.8	1.3 ± 0.5 *	1.5 ± 0.5	1.7 ± 0.9 *	1.6 ± 0.5	1.7 ± 0.9	1.7 ± 1.2	1.7 ± 1.6
Marital status Married Single, divorced or separated Widowed	60 (29.1) 58 (28.2) 88 (42.7)	14 (40.0) 9 (25.7) 12 (34.3)	15 (29.4) 15 (29.4) 21 (41.2)	59 (31.1) 52 (27.4) 79 (41.6)	18 (22.5) 19 (23.8) 43 (53.8)	56 (34.8) 48 (29.8) 57 (35.4)	29 (31.9) 22 (24.2) 40 (43.9)	45 (30.0) 45 (30.0) 60 (40.0)
Living arrangements/living alone No Yes	157 (76.2) 49 (23.8)	27 (77.1) 8 (22.9)	62 (77.5) 18 (22.5)	122 (77.5) 39 (24.2)	62 (77.5) 18 (22.5)	122 (75.8) 39 (24.2)	66 (72.5) 25 (27.5)	118 (78.7) 32 (21.3)
Education/years of schooling <10 years ≥10 years	122 (58.2) 13 (37.4)	22 (62.9) 84 (40.8)	27 (52.9) 24 (47.1)	117 (61.6) 73 (38.4)	56 (70.0) 24 (30.0)	88 (54.7) 73 (45.3) *	56 (61.5) 35 (38.5)	88 (58.7) 62 (41.3)
Income source Pension No Yes	65 (31.5) 141 (68.5)	9 (25,7) 26 (74.3)	23 (45.1) 28 (54.9)	51 (26.8) 139 (73.2) *	23 (28.8) 57 (71.3)	51(31.7) 110 (68.3)	29 (31.9) 62 (68.1)	45 (30.0) 105 (70.0)
Medical services None Government Private or other	8 (3.9) 187 (90.8) 11 (5.3)	1 (2.9) 32 (91.5) 2 (5.7)	1 (1.9) 172 (90.5) 10 (5.3)	8 (4.2) 47 (92.2) 3 (5.9)	2 (2.5) 73 (91.3) 5 (6.3)	7 (4.3) 146 (90.7) 8 (4.9)	1 (1.1) 85 (93.4) 5 (5.5)	8 (5.3) 134 (89.3) 8 (5.3)
Self-perception of health Bad Good	27 (13.1) 179 (86.9)	8 (22.9) 27 (77.1)	11 (21.6) 49 (78.4)	24 (12.6) 166 (87.4)	19 (23.8) 61 (76.3)	16 (9.9) 145 (90.1) *	19 (20.9) 72 (79.1)	16 (10.7) 134 (89.3) *
Depression symptoms/CESD-7 ^e^ No, ≤4 CESD-7 scale Yes, ≥5 CESD-7 scale	120 (58.3) 86 (41.8)	19 (54.3) 16 (45.7)	31 (60.8) 20 (39.2)	108 (56.8) 82 (43.2)	36 (45.0) 44 (55.0)	103 (63.9) 58 (36.0) *	54 (59.3) 37 (40.7)	85 (56.7) 56 (43.3)
Comorbidity No, ≤2 diseases Yes, ≥3 diseases	153 (74.3) 53 (25.7)	23 (65.7) 12 (34.3)	32 (62.7) 19 (37.3)	144 (75.8) 46 (24.2)	57 (71.3) 23 (28.8)	119 (73.9) 42 (26.1)	65 (71.4) 26 (28.6)	111 (74.9) 39 (26.0)
Cognitive impairment/MMSE ^f^ No Yes	181 (87.9) 25 (12.1)	31 (88.6) 4 (11.4)	45 (88.2) 6 (11.8)	167 (87.9) 23 (12.1)	59 (73.8) 21 (26.3)	153 (95.0) 8 (4.9) *	77 (84.6) 14 (15.4)	135 (90.0) 15 (10.0)
Other diseases Osteoarthritis No Yes	148 (72.2) 57 (27.8)	22 (64.7) 12 (35.3)	38 (76.0) 12 (24.0)	132 (69.8) 57 (30.2)	54 (69.2) 24 (30.8)	116 (72.1) 45 (27.9)	62 (68.9) 28 (31.1)	109 (72.5) 41 (27.5)
Alcohol consumption <2 glasses/day >2 glasses/day	110 (53.4) 96 (46.6)	17 (50.0) 17 (50.0)	26 (50.9) 25 (49.0)	101 (53.3) 88 (46.6)	49 (61.3) 31 (38.8)	78 (48.8) 82 (51.3)	47 (51.7) 44 (48.4)	80 (53.7) 69 (46.3)
Smoking No Yes	187 (90.8) 19 (9.2)	33 (94.3) 2 (5.7)	49 (96.1) 2 (3.9)	171 (90.0) 19 (10.0)	76 (95.0) 4 (5.0)	144 (89.4) 17 (10.6)	87 (95.6) 4 (4.4)	133 (88.7) 17 (11.3) *
Polypharmacy No, ≤4 drugs/day Yes, ≥5 drugs/day	102 (49.5) 104 (50.5)	15 (42.9) 20 (57.1)	25 (49.0) 26 (50.9)	92 (48.4) 98 (51.6)	34 (42.5) 46 (57.5)	83 (51.6) 78 (48.5)	40 (43.9) 51 (56.0)	77 (51.3) 73 (48.7)
Oral health/dry mouth No Yes	179 (86.8) 27 (13.2)	26 (74.3) 9 (25.7) *	41 (80.4) 10 (19.6)	163 (86.2) 26 (13.8)	64 (79.8) 16 (20.3)	141 (87.6) 20 (12.4)	78 (85.7) 13 (14.3)	126 (84.6) 23 (15.4)
Nutritional status MNA score ^g^	25.3 ± 2.7	23.8 ± 3.4 *	24.1 ± 2.9	25.3 ± 2.7*	24.2 ± 2.9	25.5 ± 2.7 *	24.4 ± 2.78	25.5 ± 2.8 *
Dependency/BADL ^h^ No Yes	177 (85.9) 29 (14.1)	30 (85.7) 5 (14.3)	42 (82.4) 9 (17.7)	165 (86.8) 25 (13.2)	60 (75.0) 20 (25.0)	147 (91.3) 14 (8.7)*	75 (82.4) 16 (17.6)	132 (88.0) 18 (12.0)
Dependency IADL ^i^ No Yes	141 (68.5) 65 (31.6)	22 (62.9) 13 (37.1)	34 (66.7) 17 (33.3)	129 (67.9) 61 (32.1)	35 (43.8) 45 (56.3)	128 (79.5) 33 (20.5) *	52 (57.1) 39 (42.9)	111 (74.0) 39 (26.9) *
HGS, kg	14.4 ± 6.4	13.1 ± 6.2	13.4 ± 6.0	14.4 ± 6.4	12.8 ± 6.2	14.8 ± 6.3 *	8.9 ± 3.8	17.4 ± 5.2
GS, m/s	0.88 ± 0.0	0.88 ± 0.0	0.88 ± 0.0	0.89 ± 0.0	0.87 ± 0.0	0.89 ± 0.0 *	0.88 ± 0.0	0.89 ± 0.0

Results are reported as mean values ± SD, and numbers and percentages. ^a^ Fat mass; Excess fat mass by FMI classification, ≥6.0 kg/m^2^ in men, ≥9.0 kg/m^2^ in women. ^b^ Low total lean tissue by TLTI classification, ≤15.6 kg/m^2^ in men, ≤13.5 kg/m^2^ in women. ^c^ Low gait speed ≤0.8 m/s in men, ≤0.7 m/s in women. ^d^ Low handgrip strength, ≤18 kg in men, ≤10 kg in women. DXA, dual-energy X-ray absorptiometry. ^e^ CESD-7, Center for Epidemiologic Studies Depression Scale. ^f^ MMSE, Mini Mental State Examination, ≤23 points were obtained with 5 years of school education, ≤19 points with between 1 and 4 years of education, ≤16 without education, or with <1 year of education. ^g^ MNA, mini nutritional assessment. ^h^ Dependency based on the BADL, basic activities of daily living, ≤90 points on the Barthel index. ^i^ Dependency based on IADL, instrumental activities of daily living <8 points on the Lawton and Brody scale. HGS, handgrip strength. GS, gait speed. BMI, body mass index. BMC, bone mineral content. * *p* ≤ 0.05.

**Table 2 nutrients-15-04419-t002:** Cumulative incidence of RM in community-dwelling older adults at 4.1 years of follow-up.

Variables	Incidence of RM		*p*-Value
	Yes (*n* = 69)	No (*n* = 172)	
Fat mass Excess (*n* = 206) Normal (*n* = 35)	52 (25.2) 17 (48.5)	154 (74.8) 18 (51.5)	0.004
Total lean tissue Low (*n*= 51) Normal (*n* = 190)	22 (41.1) 47 (24.7)	29 (58.9) 143 (75.3)	0.009
Gait speed Low (*n* = 80) Normal (*n* = 161)	33 (41.2) 36 (23.8)	47 (58.8) 125 (76.2)	0.002
Handgrip strength Low (*n* = 91) Normal (*n* = 150)	33 (36.2) 36 (24.0)	52 (63.8) 114 (76)	0.041

Fat mass; excess fat mass by FMI classification, ≥6.0 kg/m^2^ in men, ≥9.0 kg/m^2^ in women. Total lean tissue; Low TLT by TLTI classification, ≤15.6 kg/m^2^ in men, ≤13.5 kg/m^2^ in women. Low gait speed ≤0.8 m/s in men, ≤0.7 m/s in women. Low handgrip strength, ≤18 kg in men, ≤10 kg in women.

**Table 3 nutrients-15-04419-t003:** Relative changes at 4.1 years of follow-up in several variables in community-dwelling older adults.

Variables	Baseline	Follow-Up	*p*-Value	∆
Age, years	74.3 ± 7.6	78.6 ± 7.7	0.000	5.0
Anthropometry				
Height, m	1.5 ± 0.0	1.5 ± 0.1	0.028	−9.5
Body weight, kg	64.7 ± 12.6	62.1 ± 12.8	0.000	−4.1
Body mass index, kg/m^2^	28.5 ± 4.9	27.5 ± 5.0	0.000	−3.6
Body composition				
Total mass by DXA, kg	62.2 ± 12.8	60.6 ± 8.2	0.009	−2.6
Fat mass, kg	26.2 ± 7.7	24.8 ± 7.9	0.000	−5.6
Fat mass index, kg/m^2^	11.6 ± 3.3	11.0 ± 3.3	0.000	−5.4
Total lean tissue, kg	34.4 ± 6.4	34.1 ± 6.3	0.601	−1.0
Total lean tissue index, kg/m^2^	15.2 ± 2.2	15.1 ± 2.2	0.826	−0.6
Bone mineral content, kg	1.7 ± 0.3	1.7 ± 0.5	0.087	−1.1
Nutritional status				
MNA, score	25.5 ± 2.8	23.0 ± 3.1	0.000	−10.8
Physical performance				
HGS, kg	14.6 ± 6.3	13.7 ± 5.9	0.064	−6.5
GS, m/s	0.9 ± 0.0	0.8 ± 0.0	0.012	−12.5

Results are reported as mean values ± SD. DXA, dual-energy X-ray absorptiometry. MNA, mini nutritional assessment. HGS, handgrip strength. GS, gait speed.

**Table 4 nutrients-15-04419-t004:** Association between the hypothesis variables and several independent variables with RM by simple logistic regression analysis in community-dwelling older adults.

Variables	OR	*p*-Value	CI 95%
Excess fat mass, kg/m^2^	0.35	0.006	0.17–0.74
Low TLT, kg/m^2^	2.30	0.011	1.21–4.39
Gender, women	1.06	0.862	0.50–2.28
Age, years	1.01	0.274	0.98–1.05
Marital status Single, divorced, or separated Widowed	0.55 0.93	0.125 0.841	0.25–1.18 0.49–1.78
Living alone	0.67	0.267	0.33–1.35
Education, <10 years of schooling	1.98	0.025	0.27–0.91
Without a pension	1.23	0.500	0.66–2.29
Without medical services	1.35	0.674	0.32–5.59
Good self-perception of health	0.15	0.000	0.06–0.32
Depression symptoms, ≥5 CESD-7 scale	1.48	0.168	0.84–2.60
Cognitive impairment	1.91	0.110	0.86–4.26
Comorbidity, ≥3 diseases	1.87	0.042	1.02–3.43
Osteoarthritis	0.93	0.842	0.50–1.75
Polypharmacy, ≥5 drugs/day	1.57	0.118	0.89–2.76
Dry mouth	3.95	0.000	1.90–8.21
Dependency by the BADL	6.24	0.000	2.87–13.54
Dependency by the IADL	2.82	0.000	1.57–5.06
Low gait speed, m/s	1.96	0.020	1.11–3.45
Low handgrip strength, kg	1.80	0.042	1.02–3.18
Alcohol consumption, >2 glasses/day	1.41	0.547	0.45–4.38
Smoking	0.76	0.610	0.26–2.16

Excess fat mass by FMI classification ≥6.0 kg/m^2^ in men, ≥9.0 kg/m^2^ in women. Low total lean tissue by TLTI classification, ≤15.6 kg/m^2^ in men, ≤13.5 kg/m^2^ in women. Low gait speed ≤0.8 m/s in men, ≤0.7 m/s in women. Low handgrip strength, ≤18 kg in men, ≤10 kg in women. CESD-7 Center for Epidemiologic Studies Depression Scale. Cognitive impairment by MMSE, mini mental state examination, ≤23 points were obtained with 5 years of school education, ≤19 points with 1–4 years of education, ≤16 without education, or with <1 year. Dependency based on the BADL, basic activities of daily living, ≤90 points on the Barthel index. Dependency based on the IADL, instrumental activities of daily living, <8 points on the Lawton and Brody scale. Low gait speed ≤0.8 m/s in men, ≤0.7 m/s in women. Low handgrip strength, ≤18 kg in men, ≤10 kg in women.

**Table 5 nutrients-15-04419-t005:** Final models of the association between alterations of body composition and impaired physical performance and RM in community-dwelling older people.

			Model 1		Model 2	
	OR Unadjusted (CI 95%)	*p*-Value	OR Adjusted (CI 95%)	*p*-Value	OR Adjusted (CI 95%)	*p*-Value
Fat mass Normal Excess	1.00 0.35 (0.17–0.74)	0.006	1.00 0.24 (0.09–0.64)	0.005	1.00 0.35 (0.16–0.82)	0.015
Total lean tissue Normal Low	1.00 2.30 (1.21–4.39)	0.011	1.00 2.06 (0.98–4.32)	0.054	1.00 2.04 (0.97–4.26)	0.057
Gait speed, m/s Normal Low	1.00 2.43 (1.36–4.32)	0.003	1.00 2.01 (1.06–3.83)	0.032	1.00 1.96 (1.03–3.72)	0.038
Handgrip strength, kg Normal Low	1.00 1.80 (1.02–3.18)	0.042	1.00 1.68 (0.90–3.16)	0.061	1.00 1.63 (0.87–3.05)	0.125

Excess fat mass by FMI classification, kg/m^2^; Model 1 adjusted for BADL, gait speed, and self-perception of health. Model 2 adjusted for IADL, gait speed, and self-perception of health. Low total lean tissue by TLTI classification, kg/m^2^; model 1 adjusted for BADL, dry mouth, and self-perception of health; model 2 adjusted for IADL, dry mouth, gait speed, and self-perception of health. Gait speed, m/s; model 1 adjusted for dry mouth, fat mass index, and self-perception of health; model 2 adjusted for dry mouth, total lean tissue index, and self-perception of health. Handgrip strength, kg; model 1 adjusted for dry mouth, fat mass index, and BADL; model 2 adjusted for dry mouth, total lean tissue index, and self-perception of health. Excess fat mass by FMI classification ranges, fat mass index ≥6.0 kg/m^2^ in men, ≥9.0 kg/m^2^ in women. Low TLT by TLTI classification, ≤15.6 kg/m^2^ in men, ≤13.5 kg/m^2^ in women. CESD-7 Center for Epidemiologic Studies Depression Scale. Low handgrip strength, ≤18 kg in men, ≤10 kg in women. Low gait speed ≤0.8 m/s in men, ≤0.7 m/s in women. Low handgrip strength, ≤18 kg in men, ≤10 kg in women.

## Data Availability

The database used for the present analysis is available under request.

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
