# Peer review of "Incidence of the Risk of Malnutrition and Excess Fat Mass, and Gait Speed as Independent Associated Factors in Community-Dwelling Older Adults"

_nutrients, 2023, doi:10.3390/nu15204419_

Round 1

Reviewer 1 Report

This is a prospective study evaluating the factors associated with the risk of malnutrition. The paper is well-written and have some interesting results, such as showing that an excess of fat mass could be a protective effect of RM, whereas low lean tissue and low gait speed acted as a risk factor. Nonetheless, there are some flaws that the authors should take in to account for making this publication suitable for publication.

Q1: I would suggest considering the inclusion of a paragraph in the study design section, around line 132, rather than within the study population section.

Q2: It's worth noting that RM has also been associated with cognitive impairment.

Q3: In line 158, there appears to be a potential mistake in the cutoff point for the risk of malnutrition. The way it is currently written suggests that people with values between 17 and 23.5 are excluded from the analysis.

Q4: There are discrepancies in Table 1. When the numbers are summed, the total is 482 participants, which does not align with either the initial baseline number or the final study number. I strongly recommend including data only from participants who completed the study, rather than the original baseline data from another study. Additionally, the results section from line 315 to 330 is based on this data, which does not match the number of participants.

Q5: It would be valuable to include a descriptive table with data on the population at baseline and after the follow-up to assess how other variables changed during that period.

Q6: Given the extensive number of variables in this study, have the authors considered conducting a PCA analysis to facilitate statistical analysis?

Q7: In the flowchart, it appears that participants younger than 60 years are included, which seems inconsistent with the inclusion criteria for this study.

Q8: Table 3 would benefit from additional information, such as confidence intervals, to provide a more visual representation of the data.

Q9: Marital status, depression, cognitive impairment, and polypharmacy are mentioned as risk factors for RM, but the table does not seem to support these associations.

Q10: It's advisable to include age as a factor in the analysis of Table 3 to explore its potential influence.

Q11: The data presented in lines 385 to 390 do not align with the findings in Table 4. Some significant differences are mentioned as non-significant in Table 4.

Q12: Line 413 could be confusing since all subjects with RM at baseline were excluded. Consider rephrasing for clarity.

Q13: Given that higher fat mass is associated with a protective effect against the risk of malnutrition in this paper, it may be appropriate to remove the "however" from line 413.

Q14: The authors should consider providing a more comprehensive discussion of the associations between gait speed and RM, including comparisons with other studies (e.g., https://doi.org/10.1093%2Fcdn%2Fnzac007) and discussing the implications of this association.

Q15: Expanding the discussion with a broader analytical framework would enhance the depth of the paper's analysis.

Q16: The references in line 105 need to be properly formatted and cited.

The English quality is good; however, a grammatical and structured revision could improve the readiness of the publication.

Author Response

See attach document

Reviewer 2 Report

  1. 1. The authors appear to misinterpret their results. It is essential to emphasize that if variables are found to be non-significant in the multivariate models, this should be the focus of the conclusion rather than dwelling on the results of univariate analyses. The authors should make it clear that non-significant variables in the multivariate analysis do not contribute significantly to the outcomes of interest and avoid discussing them extensively. It is crucial for the clarity and accuracy of the paper.

  2. 2. The authors should provide a more comprehensive argument for the cut-off values they have chosen. Additionally, conducting sensitivity analyses with alternative threshold values would enhance the robustness of their findings. This would help assess the stability of their results and provide readers with a better understanding of the potential impact of different cut-off values on the study outcomes.

  3. 3. The presentation of tables in the manuscript needs improvement. Some tables lack clarity, and there are issues with overlapping titles. Clear and concise table formatting is essential for readers to comprehend the results effectively. The authors should ensure that each table is well-structured, and the titles are appropriately placed to avoid confusion.

Author Response

See attach document

Reviewer 3 Report

This is a high-impact manuscript that identifies factors correlated with RM through a large, long-term cohort study. But some modifications are required for publication in Nutrients.

The keywords "80 and over" are inappropriate considering the average age of the subjects in this study.

Line151: It is very important in this survey what questions were asked about “food and fluid intake”. It is important to clearly describe what questions were asked for participants.

Line 247: What is the ethanol volume equivalent?

Statistical analysis: Although the step wise method is selected, the forced entry method may be more appropriate since there are few previous studies in this field. So, result of this should be appended in supplementary file.

Table 1 is confusing and difficult to read. I could not understand where the “normal” and “low” of fat mass index, total lean tissue, gait speed, and hand grip strength are listed respectively in the table. Why is there no EXCESS other than Fat mass index?

Table 3 95% confidence intervals should be listed.

Author Response

See attach document

Round 2

Reviewer 1 Report

The article has improved, congratulations to the authors. 

Author Response

Thanks

Reviewer 2 Report

The authors still overinterpret their results.  Sentences such as "A strong tendency toward significance" is far from good sciences! As previously stated, the authors must focus on significant results after multivariate analyses.

Reviewer 3 Report

I think authors have corrected well on the points I suggested.

Author Response

Thanks